

SciPost Phys. 1(1), 003 (2016)

# Time evolution during and after finite-time quantum quenches in the transverse-field Ising chain

**T. Puškarov**, **D. Schuricht**

Institute for Theoretical Physics, Center for Extreme Matter and Emergent Phenomena,
Utrecht University, Princetonplein 5, 3584 CC Utrecht, the Netherlands

t.puskarov@uu.nl, d.schuricht@uu.nl

## Abstract

We study the time evolution in the transverse-field Ising chain subject to quantum quenches of finite duration, ie, a continuous change in the transverse magnetic field over a finite time. Specifically, we consider the dynamics of the total energy, one- and two-point correlation functions and Loschmidt echo during and after the quench as well as their stationary behaviour at late times. We investigate how different quench protocols affect the dynamics and identify universal properties of the relaxation.

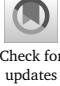
# 1  Introduction

Experiments on ultracold atomic gases in optical lattices have opened the opportunity to simulate condensed-matter models in a well-controlled setup [1–3], and also to probe the non-equilibrium dynamics of such strongly correlated quantum systems [4–9]. These systems are isolated from their environment to the extent that we can consider them closed, and they allow for dynamic control of the system parameters, enabling investigation of dynamics induced by quantum quenches [10, 11], ie, the time evolution following a sudden change in one of the system parameters. This triggered a considerable interest in theoretically addressing the non-equilibrium behaviour of various condensed-matter models as a consequence of sudden quenches, see References [12–14] for a review. A natural extension of this setting are finite-time quenches, ie, the study of the dynamics during and after the change of the system parameters over a finite time interval $\tau$. Obviously the two extreme limits to this problem are the sudden quench and the adiabatically slow change of parameters, but the huge regime between these limits and the generic time dependence of the parameters open many possibilities for new features in the non-equilibrium dynamics. The main aim of the present manuscript is the study of this dynamics in the prototypical transverse-field Ising chain. So far finite-time quenches have been investigated mostly in bosonic as well as fermionic Hubbard models [15–17] and one-dimensional Luttinger liquids [18–25].

In this work, we focus on finite-time quenches in the transverse field Ising (TFI) model. The Ising model has been realised experimentally in an ultracold gas of bosonic atoms in a linear potential [26], and its behaviour following sudden quenches across the critical point has been observed [27]. Theoretically, going back to the 70s [28–30] sudden quenches have been studied extensively in this system. In particular, the order parameter and spin correlation functions [31, 32] as well as the generalised Gibbs ensemble [33–35] describing the late time stationary state have been investigated in detail. For example, as a consequence of the Lieb–Robinson bounds [36, 37] the spin correlation functions show a clear light-cone effect [10]. In our results, those behaviours are reproduced and generalised to take into account the finite-quench duration and non-sudden protocol. We note that in the context of the Kibble–Zurek mechanism some attention has been given to linear ramps through the quantum phase transition in the TFI model [38–40]. Furthermore, the behaviour of the order parameter in the late-time limit after linear ramps in the ferromagnetic phase has been investigated [41].

This paper is organised as follows: In section 2 we set the notation and review the diagonalisation of the TFI chain. In section 3 we discuss the setup of a finite-time quantum quench and derive expressions for the time evolution of the correlation functions in the TFI chain without addressing the specifics of the quench protocol. We also construct the generalised Gibbs ensemble describing the stationary state at late times after the quench, again without addressing specific quench protocols. In section 4 we discuss specific quench protocols: linear, exponential, cosine and sine, cubic and quartic quenches, as well as piecewise

differentiable versions thereof. Hereby the protocols are chosen to cover different features of the time dependence like non- differentiable kinks. This allows us to identify properties of the non-equilibrium dynamics that are universal, ie, independent of these details. We derive the equations governing the time-dependent Bogoliubov coefficients for each of the protocols and calculate their exact solutions for the linear and exponential quenches. In section 5 we analyse the behaviour of the total energy, transverse magnetisation, transverse and longitudinal spin correlation functions, and the Loschmidt echo during and after the quench. In section 6 we briefly discuss the scaling limit of our results. Finally, in section 7 we re-interpret our results in the context of time-dependently curved spacetimes [42] before concluding with an outlook in section 8.

## 2 Transverse field Ising (TFI) chain

The Hamiltonian of the $N$-site TFI chain is given by

$$H = -J \sum_{i=1}^{N} \left( \sigma_i^x \sigma_{i+1}^x + g \sigma_i^z \right), \tag{1}$$

where $\sigma^a$, $a = x, y, z$, are the Pauli matrices, $J > 0$ sets the energy scale and periodic boundary conditions $\sigma_{N+1}^a = \sigma_1^a$ are imposed. The dimensionless parameter $g$ describes the coupling to an external, transverse magnetic field. In the thermodynamic limit, the TFI chain at zero temperature is a prototype system which exhibits a quantum phase transition [43]. The transition occurs between the ferromagnetic (ordered) phase for $g < 1$ and the paramagnetic (disordered) phase for $g > 1$, with the critical point being $g_c = 1$.

The Hamiltonian (1) can be exactly diagonalised by transforming to a spinless representation [32, 44, 45]. First, using the spin raising and lowering operators $\sigma_i^\pm = \frac{1}{2} \left( \sigma_i^x \pm i \sigma_i^y \right)$ it is recast into

$$H = -J \sum_{i=1}^{N} \left( \sigma_i^+ + \sigma_i^- \right) \left( \sigma_{i+1}^+ + \sigma_{i+1}^- \right) - J g \sum_{i=1}^{N} \left( 1 - 2 \sigma_i^+ \sigma_i^- \right). \tag{2}$$

We can then transform the spin operators to fermions by means of a Jordan–Wigner transformation

$$c_i = \exp\left( i\pi \sum_{j<i} \sigma_j^+ \sigma_j^- \right) \sigma_i^-, \quad c_i^\dagger = \sigma_i^+ \exp\left( i\pi \sum_{j<i} \sigma_j^+ \sigma_j^- \right), \tag{3}$$

where $c_i$ and $c_i^\dagger$ are spinless fermionic creation and annihilation operators at lattice site $i$. The Hamiltonian in terms of the Jordan–Wigner fermions obtains a block-diagonal structure $H = H_e \oplus H_o$, where

$$H_{e/o} = -J \sum_{i=1}^{N} \left( c_i^\dagger - c_i \right) \left( c_{i+1}^\dagger + c_{i+1} \right) - J g \sum_{i=1}^{N} \left( 1 - 2 c_i^\dagger c_i \right). \tag{4}$$

The reduced Hamiltonian $H_{e/o}$ acts only on the subspace of the Fock space with even/odd number of fermions. In the sector with an even fermion number, the so-called Neveu–Schwarz (NS) sector, the fact that $\exp(i\pi \sum_{i=1}^{N} c_i^\dagger c_i) = 1$ implies that the fermions have to satisfy antiperiodic boundary conditions $c_{N+1} = -c_1$. Similarly, in the sector with odd fermion number, usually referred to as Ramond (R) sector, the relation $\exp(i\pi \sum_{i=1}^{N} c_i^\dagger c_i) = -1$ implies periodic boundary conditions $c_{N+1} = c_1$.

We perform a discrete Fourier transformation to momentum space as $c_k = \frac{1}{\sqrt{N}} \sum_{i=1}^{N} c_i e^{iki}$, where we have set the lattice spacing to unity. The Hamiltonian becomes

$$H_{e/o} = -J g N + 2J \sum_k (g - \cos k) c_k^\dagger c_k - i J \sum_k \sin k \left( c_{-k}^\dagger c_k^\dagger + c_{-k} c_k \right), \tag{5}$$

where the sum over momenta $k$ implies the sum over $n = -\frac{N}{2}, \ldots, \frac{N}{2} - 1$, and the momenta are quantised as $k_n^{\mathrm{NS}} = \frac{2\pi}{N}(n + \frac{1}{2})$ in the even and $k_n^{\mathrm{R}} = \frac{2\pi n}{N}$ in the odd sector.

In the even sector, the Hamiltonian can finally be diagonalised by applying the Bogoliubov transformation

$$\eta_k = u_k c_k - \mathrm{i} v_k c_{-k}^\dagger, \qquad \eta_k^\dagger = u_k c_k^\dagger + \mathrm{i} v_k c_{-k}. \tag{6}$$

We choose the transformation such that the Bogoliubov coefficients $u_k$ and $v_k$ are real. From the requirement that the Bogoliubov operators satisfy the usual fermionic anticommutation relations, we obtain $u_k^2 + v_k^2 = 1$ as well as $u_k = u_{-k}$ and $v_k = -v_{-k}$. We can therefore parametrise the Bogoliubov coefficients as $u_k = \cos\frac{\theta_k}{2}$ and $v_k = \sin\frac{\theta_k}{2}$. The requirement in the new representation is that the off-diagonal terms of the Hamiltonian vanish, which yields the condition

$$\mathrm{e}^{\mathrm{i}\theta_k} = \frac{g - \mathrm{e}^{\mathrm{i}k}}{\sqrt{1 + g^2 - 2g\cos k}}. \tag{7}$$

The Hamiltonian is then diagonalised as $H_{\mathrm{e}} = \sum_k \varepsilon_k \left( \eta_k^\dagger \eta_k - \frac{1}{2} \right)$, where the single particle dispersion relation is $\varepsilon_k = 2J\sqrt{1 + g^2 - 2g\cos k}$. The excitation gap is thus given by $\Delta = \varepsilon_{k=0} = 2J|1 - g|$.

In the odd sector, the diagonalisation proceeds similarly using the Bogoliubov transformation (6). Additional care has to be taken for the momenta $k_0 = 0$ and $k_{-N/2} = -\pi$, which do not have partners with $-k$. The resulting Hamiltonian is $H_{\mathrm{o}} = \sum_{k \neq 0} \varepsilon_k \left( \eta_k^\dagger \eta_k - \frac{1}{2} \right) - 2J(1 - g) \times \left( \eta_0^\dagger \eta_0 - \frac{1}{2} \right)$.

## 3 Finite-time quantum quenches

### 3.1 General quench protocols

In the setup we consider, the system is initially prepared in the ground state of the Hamiltonian $H(g_{\mathrm{i}})$, which is the vacuum state for the Bogoliubov fermions $\eta_k$ and $\eta_k^\dagger$. Starting at $t = 0$ the coupling to the transverse field is continuously changed over a finite quench time $\tau$ until it reaches its final value $g_{\mathrm{f}}$. Following the quench, ie, for times $t > \tau$, the system evolves according to the Hamiltonian $H(g_{\mathrm{f}})$. In other words, we consider the time-dependent system

$$H(t) = -J \sum_{i=1}^{N} \left( \sigma_i^x \sigma_{i+1}^x + g(t)\sigma_i^z \right), \tag{8}$$

with the continuous function $g(t)$ taking the limiting values

$$g(t < 0) = g_{\mathrm{i}}, \quad g(t > \tau) = g_{\mathrm{f}}. \tag{9}$$

Some examples of protocols $g(t)$ are shown in Fig. 1.

We are interested in the dynamical behaviour of physical observables, ie, in calculating the expectation values of time-evolved operators taken with respect to the initial ground state. Unlike in the sudden-quench case where the pre- and post-quench Bogoliubov fermions are directly related [46], for general protocols one has different instantaneous Bogoliubov fermions and Bogoliubov coefficients at each time, which is not practical. Instead, we assume the following ansatz for the time evolution of the Jordan–Wigner fermions [38]

$$c_k(t) = u_k(t)\eta_k + \mathrm{i} v_k(t)\eta_{-k}^\dagger, \tag{10}$$

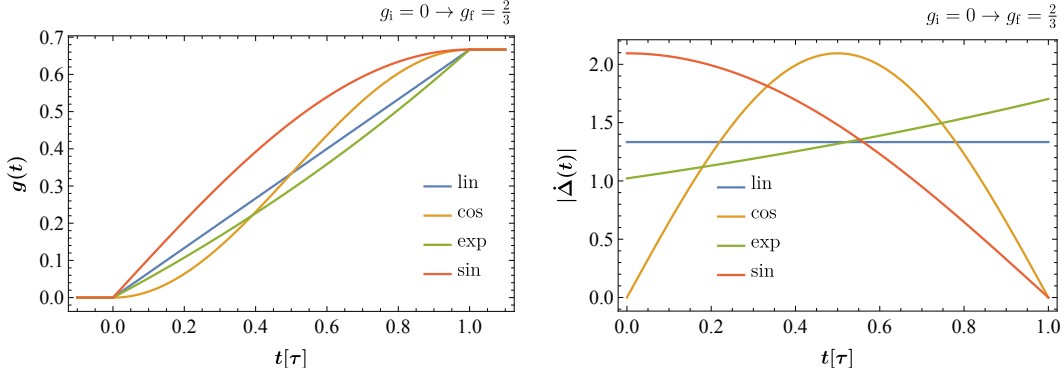

Figure 1: Left: Sketch of different quench protocols for $g_i = 0$ to $g_f = \frac{2}{3}$. Right: Corresponding change rate of the gaps.

ie, we keep the Bogoliubov fermions $\eta_k$ which diagonalise the initial Hamiltonian and cast the temporal dependence into the functions $u_k(t)$ and $v_k(t)$. Making use of the Heisenberg equations of motion for the operators $c_k(t)$ and $c_k^\dagger(t)$ we obtain

$$i\frac{d}{dt}\begin{pmatrix} u_k(t) \\ v_{-k}^*(t) \end{pmatrix} = \begin{pmatrix} A_k(t) & B_k \\ B_k & -A_k(t) \end{pmatrix}\begin{pmatrix} u_k(t) \\ v_{-k}^*(t) \end{pmatrix}, \tag{11}$$

with $A_k(t) = 2J[g(t) - \cos k]$, $B_k = 2J \sin k$, and the asterisk $*$ denoting complex conjugation. According to (6) the initial conditions read

$$u_k(t=0) = \cos\frac{\theta_k^i}{2}, \quad v_k(t=0) = \sin\frac{\theta_k^i}{2}, \tag{12}$$

with the angle $\theta_k^i$ defined by (7) with the initial value $g = g_i$. The equations (11) can also be decoupled as

$$\frac{\partial^2}{\partial t^2} y_k(t) + \left(A_k(t)^2 + B_k^2 \pm i\frac{\partial}{\partial t}A_k(t)\right) y_k(t) = 0, \tag{13}$$

where the upper and lower sign refers to the equation for $y_k(t) = u_k(t)$ and $y_k(t) = v_{-k}^*(t)$ respectively. During the quench, the solutions to these equations depend on the precise form of $g(t)$ and in some cases allow for explicit analytic solutions. We will address several of these protocols in section 4.

After the quench, the equations for the Bogoliubov coefficients simplify to

$$\frac{\partial^2}{\partial t^2} y_k(t) + \omega_k^2 y_k(t) = 0, \tag{14}$$

with the solution

$$y_k(t) = c_3^y e^{i\omega_k t} + c_4^y e^{-i\omega_k t}, \quad \omega_k = \sqrt{A_k(\tau)^2 + B_k^2} = \varepsilon_k(g_f), \tag{15}$$

where we have defined the single-mode energies $\omega_k$ after the quench. The constants $c_3^y$ and $c_4^y$ are determined by the continuity of the solutions at $t = \tau$ with the results

$$c_3^u = \frac{e^{-i\omega_k\tau}}{2\omega_k}\left[\left(\omega_k - A_k(\tau)\right)u_k(\tau) - B_k v_{-k}^*(\tau)\right], \tag{16}$$

$$c_4^u = \frac{e^{i\omega_k\tau}}{2\omega_k}\left[\left(\omega_k + A_k(\tau)\right)u_k(\tau) + B_k v_{-k}^*(\tau)\right], \tag{17}$$

for $u_k(t)$, and

$$c_3^v = \frac{e^{-i\omega_k\tau}}{2\omega_k}\left[\left(\omega_k + A_k(\tau)\right)v_{-k}^*(\tau) - B_k u_k(\tau)\right], \tag{18}$$

$$c_4^v = \frac{e^{i\omega_k\tau}}{2\omega_k}\left[\left(\omega_k - A_k(\tau)\right)v_{-k}^*(\tau) + B_k u_k(\tau)\right], \tag{19}$$

for $v_{-k}^*(t)$. We stress that these constants depend on the momenta $k$ and the quench duration $\tau$, but we use the shorthand notation $c_n^y = c_{n,k}^y(\tau)$ for brevity.

## 3.2 Transverse magnetisation and correlation functions

In order to probe the system, we aim at calculating local observables during and after a time-dependent quench. The observables we have in mind are the transverse magnetisation and two-point functions in the transverse and longitudinal direction. Here we briefly sketch how to express these observables in terms of the time-dependent Bogoliubov coefficients $u_k(t)$ and $v_k(t)$.

Firstly, we write the correlators in terms of Jordan–Wigner fermions in (3) and define auxiliary operators $a_i = c_i^\dagger + c_i$ and $b_i = c_i^\dagger - c_i$ to obtain

$$M^z = \left\langle \sigma_i^z \right\rangle = \left\langle b_i a_i \right\rangle, \tag{20}$$

$$\rho_n^z = \left\langle \sigma_i^z \sigma_{i+n}^z \right\rangle = \left\langle a_i b_i a_{i+n} b_{i+n} \right\rangle, \tag{21}$$

$$\rho_n^x = \left\langle \sigma_i^x \sigma_{i+n}^x \right\rangle = \left\langle b_i a_{i+1} b_{i+1} \ldots a_{i+n-1} b_{i+n-1} a_{i+n} \right\rangle. \tag{22}$$

Here we have used $1 - 2c_i^\dagger c_i = a_i b_i = -b_i a_i$ and suppressed the time dependence of the operators for concise notation. Furthermore, due to translational invariance the observables do not depend on the lattice site. Secondly, we define the contractions, vacuum expectation values of pairs of operators, as $S_{ij} = \left\langle b_i b_j \right\rangle$, $Q_{ij} = \left\langle a_i a_j \right\rangle$ and $G_{ij} = \left\langle b_i a_j \right\rangle = -\left\langle a_j b_i \right\rangle$. The transverse magnetisation is then simply given by

$$M^z = -G_{ii}. \tag{23}$$

Employing the Wick theorem, we can express the two-point functions in terms sums of products of all possible contractions. This can be conveniently written in the form of Pfaffians [29]

$$\rho_n^z = \begin{vmatrix} S_{i,i+n} & G_{i,i} & G_{i,i+n} \\ & G_{i+n,i} & G_{i+n,i+n} \\ & & Q_{i,i+n} \end{vmatrix}, \tag{24}$$

$$\rho_n^x = \begin{vmatrix} S_{i,i+1} & S_{i,i+2} & \cdots & S_{i,i+n-1} & G_{i,i+1} & G_{i,i+2} & \cdots & G_{i,i+n} \\ & S_{i+1,i+2} & \cdots & S_{i+1,i+n-1} & G_{i+1,i+1} & G_{i+1,i+2} & \cdots & G_{i+1,i+n} \\ & & \vdots & \vdots & \vdots & \vdots & & \vdots \\ & & & S_{i+n-2,i+n-1} & G_{i+n-2,i+1} & G_{i+n-2,i+2} & \cdots & G_{i+n-2,i+n} \\ & & & & G_{i+n-1,i+1} & G_{i+n-1,i+2} & \cdots & G_{i+n-1,i+n} \\ & & & & & Q_{i+1,i+2} & \cdots & Q_{i+1,i+n} \\ & & & & & & & \vdots \\ & & & & & & & Q_{i+n-1,i+n} \end{vmatrix}. \tag{25}$$

In this triangular notation, Pfaffians can be viewed as generalised determinants which can be expanded along "rows" and "columns" in terms of minor Pfaffians [47]. They can also

be evaluated from the corresponding antisymmetric matrices $A$ as $\text{Pf}(A)^2 = \det A$. Here the matrix $A$ has vanishing elements on the main diagonal, its upper triangular part is the Pfaffian as written in (24) and (25), and the lower triangular part is the negative transpose of the Pfaffian in question. Thirdly, we introduce auxiliary quadratic correlators $\alpha_{ij}$ and $\beta_{ij}$, and express their time-dependence in terms of Bogoliubov coefficients by using (10)

$$\alpha_{ij}(t) = \left\langle c_i(t)c_j^\dagger(t) \right\rangle = \frac{1}{2\pi} \int_{-\pi}^{\pi} dk\, e^{-ik(i-j)} |u_k(t)|^2, \tag{26}$$

$$\beta_{ij}(t) = \left\langle c_i(t)c_j(t) \right\rangle = \frac{i}{2\pi} \int_{-\pi}^{\pi} dk\, e^{-ik(i-j)} u_k(t)v_{-k}(t). \tag{27}$$

Using these functions, the various contractions become

$$S_{ij}(t) = 2i\,\text{Im}(\beta_{ij}(t)) - \delta_{ij}, \tag{28}$$
$$Q_{ij}(t) = 2i\,\text{Im}(\beta_{ij}(t)) + \delta_{ij}, \tag{29}$$
$$G_{ij}(t) = 2\,\text{Re}(\beta_{ij}(t)) - 2\alpha_{ij}(t) + \delta_{ij}. \tag{30}$$

These functions are the entries of (23)–(25), which give the general expressions for the time dependence of the correlation functions. An additional simplification is the fact that we may use translational invariance of the system to write $S_{ij} = S(j-i)$, $Q_{ij} = Q(j-i)$ and $G_{ij} = G(j-i)$. It is then evident that the corresponding matrices in (24) and (25) are block-Toeplitz matrices, with entries on each descending diagonal in a block identical, which reduces the computational effort.

The behaviour of these functions depends on the form and duration of the quench and will be discussed in subsequent sections. Some general features will be treated analytically.

### 3.3 Generalised Gibbs ensemble

It is by now well established that at very late times after a sudden quench a stationary state is formed which is well described by a generalised Gibbs ensemble (GGE) [33–35, 48, 49]. This ensemble contains the infinitely many integrals of motion in the TFI chain and thus retains more information about the initial state than just its energy. Considering finite-time quenches, a similar situation appears where the role of the initial state is taken by the time-evolved state at $t = \tau$. More precisely, since the time evolution for times $t > \tau$ is governed by the time-independent Hamiltonian $H(\tau) = H(g_f)$, we can construct the GGE

$$\rho_{\text{GGE}} = \frac{1}{Z} \exp\left( -\sum_k \lambda_k n_k^f \right), \tag{31}$$

where $n_k^f = \eta_k^{f\,\dagger} \eta_k^f$ are the post-quench mode occupations with $\eta_k^{f\,\dagger}$ and $\eta_k^f$ being the Bogoliubov fermions which diagonalise $H(g_f)$. We note in passing that the mode occupations are non-local in space, but that they are related to local integrals of motion via a linear transformation [35] and can thus be used in the construction of the GGE. The Lagrange multipliers $\lambda_k$ are fixed by

$$\left\langle \eta_k^{f\,\dagger} \eta_k^f \right\rangle = \text{Tr}\left( \rho_{\text{GGE}}\, \eta_k^{f\,\dagger} \eta_k^f \right) \tag{32}$$

and $Z = \text{Tr}\left[ \exp(-\sum_k \lambda_k n_k^f) \right]$ is the normalisation. Explicitly, by first reverting to Jordan–Wigner fermions and then to the initial Bogoliubov fermions $\eta_k$, we find

$$\left\langle \eta_k^{f\,\dagger}(\tau) \eta_k^f(\tau) \right\rangle = (u_k^f)^2 |v_k(\tau)|^2 + (v_k^f)^2 |u_k(\tau)|^2 + u_k^f v_{-k}^f \left( u_{-k}(\tau)v_k(\tau) + u_{-k}^*(\tau)v_k^*(\tau) \right), \tag{33}$$

where $u_k^{\mathrm{f}}$ and $v_k^{\mathrm{f}}$ are the Bogoliubov coefficients corresponding to Bogoliubov fermions $\eta_k^{\mathrm{f}}$ as defined in (6). This allows us to fix the Lagrange multipliers by equating (33) with

$$\mathrm{Tr}\left(\rho_{\mathrm{GGE}}\,\eta_k^{\mathrm{f}\,\dagger}\eta_k^{\mathrm{f}}\right) = \frac{1}{1+\mathrm{e}^{\lambda_k}}. \tag{34}$$

We see that the Lagrange multipliers, and consequently the expectation values in the stationary state, depend on the duration $\tau$ and form $g(t)$ of the quench through the Bogoliubov coefficients $u_k(\tau)$ and $v_{-k}^*(\tau)$.

To show the validity of the GGE, we prove the equivalence of the stationary values and the GGE values of the quadratic correlators introduced in (26) and (27). Putting (15) into (26) and (27), and taking the long-time average, we obtain

$$\alpha_{ij}^{\mathrm{s}} = \frac{1}{4\pi}\int_{-\pi}^{\pi} \mathrm{d}k\, \mathrm{e}^{-ik(i-j)}\Big[1+\cos^2\theta_k^{\mathrm{f}}\big(|u_k(\tau)|^2 - |v_{-k}(\tau)|^2\big)$$
$$-\cos\theta_k^{\mathrm{f}}\sin\theta_k^{\mathrm{f}}\big(u_k(\tau)v_{-k}(\tau) + u_k^*(\tau)v_{-k}^*(\tau)\big)\Big], \tag{35}$$

$$\beta_{ij}^{\mathrm{s}} = \frac{i}{4\pi}\int_{-\pi}^{\pi} \mathrm{d}k\, \mathrm{e}^{-ik(i-j)}\Big[\sin^2\theta_k^{\mathrm{f}}\big(u_k(\tau)v_{-k}(\tau) + u_k^*(\tau)v_{-k}^*(\tau)\big)$$
$$-\cos\theta_k^{\mathrm{f}}\sin\theta_k^{\mathrm{f}}\big(|u_k(\tau)|^2 - |v_{-k}(\tau)|^2\big)\Big]. \tag{36}$$

The same result is obtained for the GGE expectation values $\alpha_{ij}^{\mathrm{GGE}} = \mathrm{Tr}\left(\rho_{\mathrm{GGE}}c_i c_j^{\dagger}\right)$ and $\beta_{ij}^{\mathrm{GGE}} = \mathrm{Tr}\left(\rho_{\mathrm{GGE}}c_i c_j\right)$.

## 4 Results for different quench protocols

In this section we collect some results for the explicit quench protocols sketched in Fig. 1, namely the linear, exponential, cosine, sine and polynomial quenches. We also define some piecewise differentiable protocols, which are later used as a check of principles but not extensively studied.

### 4.1 Linear quench

We start with the simplest finite-time quench protocol, which is the linear quench of the form (see the blue line in Fig. 1)

$$g(t) = g_{\mathrm{i}} + v_g t, \tag{37}$$

where $v_g = (g_{\mathrm{f}} - g_{\mathrm{i}})/\tau$ is the rate of change in the transverse field. For the linear quench the differential equations in (13) become

$$\partial_t^2 y_k(t) + \big[at^2 + b_k t + c_k\big]y_k(t) = 0, \tag{38}$$

with $a = 4v_g^2$, $b_k = 8v_g(g_{\mathrm{i}} - \cos k)$, $c_k = 4(1 + g_{\mathrm{i}}^2 - 2g_{\mathrm{i}}\cos k) \pm 2iv_g$, and the upper and lower signs refering to the equations for $y_k(t) = u_k(t)$ and $y_k(t) = v_{-k}^*(t)$ respectively. This equation can be cast into the form of a Weber differential equation [50] whose solutions in terms of parabolic cylinder functions are

$$y_k(t) = c_1^y D_{\nu_k}(\tilde{t}(t)) + c_2^y D_{-\nu_k-1}(i\tilde{t}(t)), \tag{39}$$

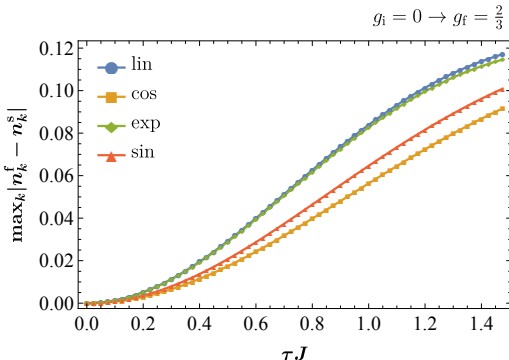

Figure 2: Comparison of the mode occupations $n_k^{\mathrm{f}}$ after different finite-time quenches with the sudden-quench result $n_k^{\mathrm{s}}$. We observe that $n_k^{\mathrm{f}} \to n_k^{\mathrm{s}}$ for $\tau \to 0$ irrespective of the quench protocol.

where $v_k = (-4\mathrm{i}ac_k + \mathrm{i}b_k^2 - 4a^{3/2})/(8a^{3/2})$ and $\tilde{t}(t) = \mathrm{e}^{\mathrm{i}\pi/4}(2^{1/2}a^{1/4}t + 2^{-1/2}a^{-3/4}b_k)$. The constants $c_1^y$ and $c_2^y$ are set by the initial conditions

$$u_k(t)|_{t=0} = u_k^{\mathrm{i}}, \tag{40}$$

$$v_{-k}^*(t)|_{t=0} = v_{-k}^{\mathrm{i}}, \tag{41}$$

$$\frac{\mathrm{d}}{\mathrm{d}t}u_k(t)|_{t=0} = -\mathrm{i}A_k(0)u_k^{\mathrm{i}} - \mathrm{i}B_k v_{-k}^{\mathrm{i}}, \tag{42}$$

$$\frac{\mathrm{d}}{\mathrm{d}t}v_{-k}^*(t)|_{t=0} = -\mathrm{i}B_k u_k^{\mathrm{i}} + \mathrm{i}A_k(0)v_{-k}^{\mathrm{i}}; \tag{43}$$

explicitly we find

$$
\begin{aligned}
c_1^u &= \frac{-D_{-\nu-1}(\mathrm{i}d_2)\left\{u_k^{\mathrm{i}}\left[2A_k(0)+\mathrm{i}d_1 d_2\right]+2v_{-k}^{\mathrm{i}}B_k\right\}+2d_1 u_k^{\mathrm{i}}D_{-\nu}(\mathrm{i}d_2)}{2d_1\left\{D_{-\nu}(\mathrm{i}d_2)D_{\nu}(d_2)+\mathrm{i}D_{-\nu-1}(\mathrm{i}d_2)\left[D_{\nu+1}(d_2)-d_2 D_{\nu}(d_2)\right]\right\}}, \\
c_2^u &= \frac{D_{\nu}(d_2)\left\{u_k^{\mathrm{i}}\left[2A_k(0)-\mathrm{i}d_1 d_2\right]+2v_{-k}^{\mathrm{i}}B_k\right\}+2\mathrm{i}d_1 u_k^{\mathrm{i}}D_{\nu+1}(d_2)}{2d_1\left\{D_{-\nu}(\mathrm{i}d_2)D_{\nu}(d_2)+\mathrm{i}D_{-\nu-1}(\mathrm{i}d_2)\left[D_{\nu+1}(d_2)-d_2 D_{\nu}(d_2)\right]\right\}},
\end{aligned}
\tag{44}
$$

for $u_k(t)$, and

$$
\begin{aligned}
c_1^v &= \frac{D_{-\nu-1}(\mathrm{i}d_2)\left\{v_{-k}^{\mathrm{i}}\left[2A_k(0)-\mathrm{i}d_1 d_2\right]-2u_k^{\mathrm{i}}B_k\right\}+2d_1 v_{-k}^{\mathrm{i}}D_{-\nu}(\mathrm{i}d_2)}{2d_1\left\{D_{-\nu}(\mathrm{i}d_2)D_{\nu}(d_2)-\mathrm{i}D_{-\nu-1}(\mathrm{i}d_2)\left[d_2 D_{\nu}(d_2)-D_{\nu+1}(d_2)\right]\right\}}, \\
c_2^v &= \frac{D_{\nu}(d_2)\left\{v_{-k}^{\mathrm{i}}\left[-2A_k(0)-\mathrm{i}d_1 d_2\right]+2u_k^{\mathrm{i}}B_k\right\}+2\mathrm{i}d_1 v_{-k}^{\mathrm{i}}D_{\nu+1}(d_2)}{2d_1\left\{D_{-\nu}(\mathrm{i}d_2)D_{\nu}(d_2)-\mathrm{i}D_{-\nu-1}(\mathrm{i}d_2)\left[d_2 D_{\nu}(d_2)-D_{\nu+1}(d_2)\right]\right\}},
\end{aligned}
\tag{45}
$$

for $v_{-k}^*(t)$. In both cases, for brevity, we use $\tilde{t}(t) = d_1 t + d_2$ with $d_1 = \mathrm{e}^{\mathrm{i}\pi/4}2^{1/2}a^{1/4}$ and $d_2 = \mathrm{e}^{\mathrm{i}\pi/4}2^{-1/2}a^{-3/4}b_k$, and we have suppressed the subindex $k$ in $v_k$ as well as in $d_2$.

In order to investigate the limit of sudden quenches we calculated the post-quench mode occupation $n_k^{\mathrm{f}}$ given in (33) after a linear quench and compared it to the post-quench mode occupation after a sudden quench $n_k^{\mathrm{s}}$. The latter is given by [46] $n_k^{\mathrm{s}} = \frac{1}{2}\left[1 - \cos(\theta_k^{\mathrm{f}} - \theta_k^{\mathrm{i}})\right]$. As is shown in Fig. 2 the difference between the two vanishes as the quench duration is decreased, ie, $\lim_{\tau \to 0} n_k^{\mathrm{f}} = n_k^{\mathrm{s}}$. As can be seen from the figure, this is true for the exponential, cosine and sine protocols as well.

## 4.2 Exponential quench

Another quench protocol which allows for an explicit analytical solution is the exponential quench of the form

$$g(t) = g_i - 1 + \exp\left(\ln(|g_f - g_i + 1|)\frac{t}{\tau}\right), \tag{46}$$

which is shown as green line in Fig. 1. The differential equations in (13) in this case become

$$\partial_t^2 y_k(t) + \left[a_k + b_k e^{ft} + c e^{2ft}\right] y_k(t) = 0, \tag{47}$$

where $a_k = 4\left[g_i^2 + 2(1 + \cos k)(1 - g_i)\right]$, $b_k = 8\left[g_i - 1 - \cos k \pm i(4\tau)^{-1}\ln|g_f - g_i + 1|\right]$, $c = 4$ and $f = \tau^{-1}\ln|g_f - g_i + 1|$. The upper and lower signs in $b_k$ refer to the equation for $y_k(t) = u_k(t)$ and $y_k(t) = v_{-k}^*(t)$ respectively. This equation can be solved using a substitution $y_k(t) = w_k(t)z_k(t)$, where $w_k(t)$ is chosen such that the equation for $z_k(t)$ reduces to an associated Laguerre equation [50]. The full solution is

$$y_k(t) = e^{i\sqrt{a_k}t}e^{i\frac{\sqrt{c}}{f}(1-e^{ft})}\left[c_1^y U(-\lambda_k, 1 + \nu_k, \tilde{t}(t)) + c_2^y L_{\lambda_k}^{\nu_k}(\tilde{t}(t))\right] \tag{48}$$

where $U(-\lambda, 1 + \nu, \tilde{t})$ denotes a confluent hypergeometric function of the second kind and $L_\lambda^\nu(\tilde{t})$ is a generalised Laguerre polynomial. Here $\lambda_k = -i\sqrt{a_k}/f - ib_k/(2f\sqrt{c}) - 1/2$, $\nu_k = 2i\sqrt{a_k}/f$ and $\tilde{t}(t) = d_1 e^{ft}$ with $d_1 = 2i\sqrt{c}/f$. The constants $c_1^y$ and $c_2^y$ are set by the initial conditions with the explicit results given by

$$c_1^u = \frac{\left[i\left(-A_k(0) - \sqrt{a} + \sqrt{c}\right)L_\lambda^\nu(d_1) + d_1 f L_{\lambda-1}^{\nu+1}(d_1)\right]u_k^i - iB_k L_\lambda^\nu(d_1)v_{-k}^i}{d_1 f\left[U(-\lambda, \nu+1, d_1)L_{\lambda-1}^{\nu+1}(d_1) + \lambda U(1-\lambda, \nu+2, d_1)L_\lambda^\nu(d_1)\right]}, \tag{49}$$

$$c_2^u = \frac{\left[i\left(A_k(0) + \sqrt{a} - \sqrt{c}\right)U(-\lambda, \nu+1, d_1) + d_1 f \lambda U(1-\lambda, \nu+2, d_1)\right]u_k^i}{d_1 f\left[U(-\lambda, \nu+1, d_1)L_{\lambda-1}^{\nu+1}(d_1) + \lambda U(1-\lambda, \nu+2, d_1)L_\lambda^\nu(d_1)\right]} \tag{50}$$
$$+ \frac{iB_k U(-\lambda, \nu+1, d_1)v_{-k}^i}{d_1 f\left[U(-\lambda, \nu+1, d_1)L_{\lambda-1}^{\nu+1}(d_1) + \lambda U(1-\lambda, \nu+2, d_1)L_\lambda^\nu(d_1)\right]},$$

$$c_1^v = \frac{\left[i\left(A_k(0) - \sqrt{a} + \sqrt{c}\right)L_\lambda^\nu(d_1) + d_1 f L_{\lambda-1}^{\nu+1}(d_1)\right]v_{-k}^i - iB_k L_\lambda^\nu(d_1)u_k^i}{d_1 f\left[U(-\lambda, \nu+1, d_1)L_{\lambda-1}^{\nu+1}(d_1) + \lambda U(1-\lambda, \nu+2, d_1)L_\lambda^\nu(d_1)\right]}, \tag{51}$$

$$c_2^v = \frac{\left[i\left(-A_k(0) + \sqrt{a} - \sqrt{c}\right)U(-\lambda, \nu+1, d_1) + d_1 f \lambda U(1-\lambda, \nu+2, d_1)\right]v_{-k}^i}{d_1 f\left[U(-\lambda, \nu+1, d_1)L_{\lambda-1}^{\nu+1}(d_1) + \lambda U(1-\lambda, \nu+2, d_1)L_\lambda^\nu(d_1)\right]} \tag{52}$$
$$+ \frac{iB_k U(-\lambda, \nu+1, d_1)u_k^i}{d_1 f\left[U(-\lambda, \nu+1, d_1)L_{\lambda-1}^{\nu+1}(d_1) + \lambda U(1-\lambda, \nu+2, d_1)L_\lambda^\nu(d_1)\right]}.$$

We stress that we have suppressed the subindex $k$ of $\nu_k$ and $\lambda_k$ for clarity.

As in the case of a linear quench, we have compared the post-quench mode occupations $n_k^f$ with the sudden-quench result (see Fig. 2). We find very similar behaviour to the linear quench even for moderate quench durations.

## 4.3 Cosine and sine quench

The cosine quench is defined as a half period of a negative cosine

$$g(t) = \frac{g_i + g_f}{2} + \frac{g_i - g_f}{2}\cos\frac{\pi t}{\tau}. \tag{53}$$

Unlike the two protocols discussed above, this protocol is differentiable for all times. Unfortunately, the differential equations (13) in this case have no analytic solution, so we have to resort to a numerical treatment.

Similarly, the sine quench is defined as a quarter period of a sine

$$g(t) = g_\text{i} + (g_\text{f} - g_\text{i}) \sin \frac{\pi t}{2\tau}. \tag{54}$$

In this protocol the transverse field initially changes faster than in the others, but slows down close to $t = \tau$. It is differentiable everywhere except at $t = 0$. Again, the differential equations (13) have no analytic solution and we study this case numerically.

The comparison of the obtained post-quench mode occupations for the cosine and sine protocols with the sudden-quench result are again shown in Fig. 2.

### 4.4 Polynomial quenches

The cubic quench is defined as

$$g = g_\text{i} - 3(g_\text{i} - g_\text{f})\left(\frac{t}{\tau}\right)^2 + 2(g_\text{i} - g_\text{f})\left(\frac{t}{\tau}\right)^3, \tag{55}$$

and the quartic quench is

$$g = g_\text{i} - 2(g_\text{i} - g_\text{f})\left(\frac{t}{\tau}\right)^2 + (g_\text{i} - g_\text{f})\left(\frac{t}{\tau}\right)^4. \tag{56}$$

Both protocols are differentiable everywhere, ie, they feature no kinks. The differential equations (13) have no analytic solution in these cases.

### 4.5 Piecewise quenches with a kink

Finally we introduce a quench protocol composed of two cosine functions stitched together at $t = \frac{\tau}{2}$ so that the protocol is continuous, but the derivative is not. The protocol is defined as

$$g(t) = \begin{cases} \frac{g_\text{f} + (1-\sqrt{2})g_\text{i}}{2-\sqrt{2}} - \frac{g_\text{f} - g_\text{i}}{2-\sqrt{2}} \cos \frac{\pi t}{2\tau}, & 0 \le t \le \frac{\tau}{2}, \\ \frac{3g_\text{f} + g_\text{i}}{4} + \frac{g_\text{f} - g_\text{i}}{4} \cos \frac{2\pi t}{\tau}, & \frac{\tau}{2} < t \le \tau. \end{cases} \tag{57}$$

Similarly, we define a protocol consisting of two linear functions with different slopes. We do this by stitching them at $t = \frac{\tau}{2}$, leading to a discontinuous derivative:

$$g(t) = \begin{cases} g_\text{i} + \frac{4}{3\tau}(g_\text{f} - g_\text{i})t, & 0 \le t \le \frac{\tau}{2}, \\ \frac{1}{3}(2g_\text{i} + g_\text{f}) + \frac{2}{3\tau}(g_\text{f} - g_\text{i})t, & \frac{\tau}{2} < t \le \tau. \end{cases} \tag{58}$$

## 5 Results for observables

### 5.1 Total energy

The simplest observable in the system is the total energy per site, $E_\text{tot} = \frac{1}{N}\langle H(t)\rangle$. Following the quench, the total energy is constant due to the unitary time evolution. During the quench, however, it depends on the quench protocol as

$$E_\text{tot} = \frac{J}{2\pi} \int_{-\pi}^{\pi} \text{d}k \left[ 2\big(g(t) - \cos k\big)|v_k(t)|^2 + \sin k \big(u_k(t)v_k(t) + u_k^*(t)v_k^*(t)\big) - g(t) \right]. \tag{59}$$

Clearly the total energy in the system depends on the quench details, as shown in 3, while it becomes independent of these details in the sudden and adiabatic limits as is well expected.

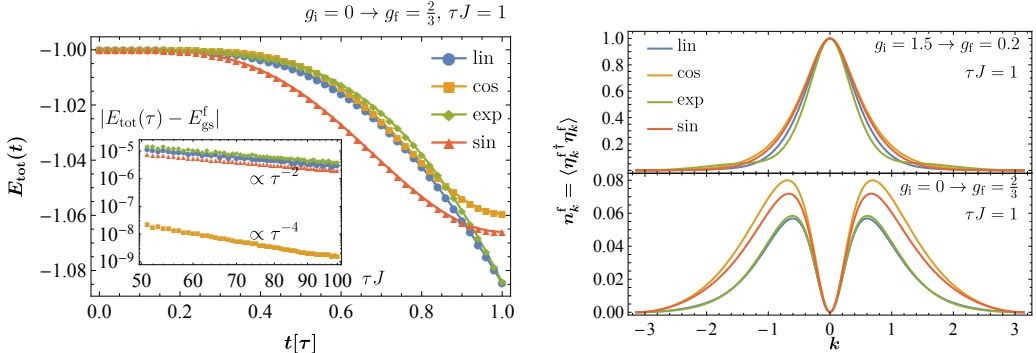

Figure 3: Left: Total energy per site $E_{\text{tot}}(t)$ during a quench from $g_{\text{i}} = 0$ to $g_{\text{f}} = \frac{2}{3}$ over $\tau J = 1$ for different quench protocols. Inset: Quenches from $g_{\text{i}} = 0$ to $g_{\text{f}} = \frac{2}{3}$ and varying durations show the approach of $E(\tau)$ to the ground-state energy of the final Hamiltonian $E_{\text{gs}}^{\text{f}}$. Right: Mode occupation $n_k^{\text{f}}$ of the final Hamiltonian at $t = \tau$ for quenches between phases (upper panel) and inside a phase (lower panel).

We find that the total energy after the quench $E_{\text{tot}}(\tau)$ approaches the adiabatic value $E_{\text{gs}}^{\text{f}}$ as a power-law, with the exponent depending on the quench details. For quenches within either the ferromagnetic or paramagnetic phase we notice two types of behaviour depending on whether the protocol has any kinks, ie, non-differentiable points: Quenches which feature kinks, ie, the linear, exponential, sine, piecewise linear and piecewise cosine quenches all approach the adiabatic value as $E_{\text{tot}}(\tau) - E_{\text{gs}}^{\text{f}} \propto \tau^{-2}$. Strikingly, quenches such as cosine, cubic and quartic, which feature no kinks, display a much faster approach, ie, $E_{\text{tot}}(\tau) - E_{\text{gs}}^{\text{f}} \propto \tau^{-4}$. The inset to Fig. 3 demonstrates the different approaches to $E_{\text{gs}}^{\text{f}}$ for several protocols. In contrast, for quenches across the critical point we find $E_{\text{tot}}(\tau) - E_{\text{gs}}^{\text{f}} \propto \tau^{-1/2}$ irrespective of the details of the protocol.

The different adiabatic behaviour for quenches between different parts of the phase diagram may be related to differences in the behaviour of the mode occupations at $k = 0$ as illustrated in Fig. 3. For quenches across the critical point (upper panel) the mode occupation at $k = 0$ is finite, while, in contrast, for quenches within a phase one finds $n_{k=0}^{\text{f}} = 0$. However, we observe no obvious difference between quench protocols with and without kinks. We note that the cosine and sine quench have a higher mode occupation, especially of the high-energy modes, and consequently a higher total energy after the quench as compared to the linear and exponential quenches as visible in the left panel.

Furthermore, at $t = \tau$ the total energy will be smooth for the cosine and sine quenches since the transverse field $g(t)$ is differentiable, while $E_{\text{tot}}$ possesses kinks for the linear and exponential quenches originating from the kinks in $g(t)$.

## 5.2 Transverse magnetisation

Next, let us now look into the behaviour of the transverse magnetisation. We can compare the magnetisation during the quench to the equilibrium ground-state magnetisation corresponding to the instantaneous value of the coupling $g(t)$, as shown in Fig. 4. If the quench duration is small compared to the inverse of the energy scales of the system, the quench is fast. In this case, the magnetisation is significantly offset from the equilibrium value for the corresponding $g(t)$. This is because the system cannot follow the change by reaching the ground state of the instantaneous Hamiltonians $H(t)$. On the other hand, as the quench slows down, the system starts its relaxation during the quench, as demonstrated by the oscillatory behaviour

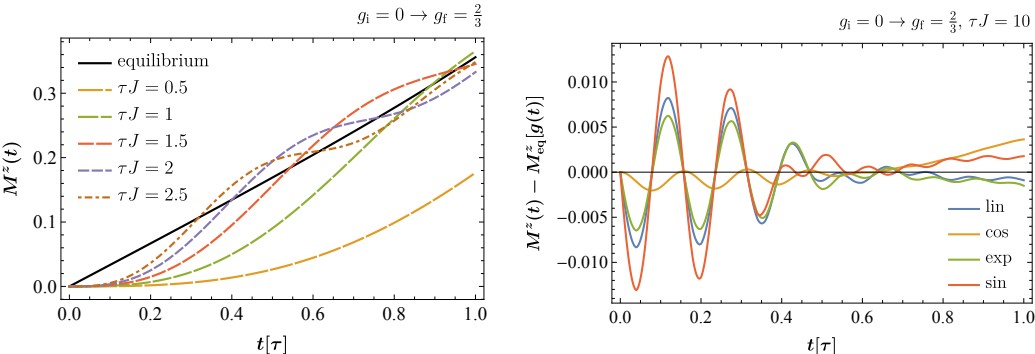

Figure 4: Transverse magnetisation during the quench for various quench durations (left) and protocols (right). Left: Linear quench from $g_i = 0$ to $g_f = \frac{2}{3}$. Full line is the equilibrium value of transverse magnetisation for a given $g(t)$. Right: Deviation of the magnetisation in linear, cosine, sine and exponential quenches from the equilibrium magnetisation for the corresponding $g(t)$.

of the magnetisation. However, in both cases there is a noticeable lag in the reaction at the very beginning of the quench. This behaviour remains qualitatively the same for different quench protocols, although there are quantitative differences, as can be seen in Fig. 4. These differences can be understood by comparing the behaviours to the gap change rates $|\dot{\Delta}|$ shown in Fig. 1. The sine quench has the highest gap change rate initially, which means that the system experiences this quench as the most violent, as demonstrated by the large amplitude of oscillations of the magnetisation from its equilibrium value. On the other hand, the cosine quench has the slowest initial gap change rate and the magnetisation in this case is much closer to the equilibrium value.

Following the quench, the magnetisation approaches a steady value. This stationary part of the magnetisation is given by $M_s^z = \lim_{t \to \infty} M^z(t)$ with the result

$$M_s^z = \frac{1}{\pi} \int_{-\pi}^{\pi} dk \left[ \cos^2 \theta_k^f \left( |v_k(\tau)|^2 - |u_k(\tau)|^2 \right) - \cos \theta_k^f \sin \theta_k^f \left( u_k(\tau) v_k(\tau) + u_k^*(\tau) v_k^*(\tau) \right) \right], \tag{60}$$

which coincides with the GGE value. The dependence of the stationary value on the duration of the quench $\tau$ and quench protocol $g(t)$ is shown in Fig. 5. In the left panel we notice that for quenches within the ferromagnetic regime an oscillatory behaviour in $\tau$ exists which is most pronounced for the linear and exponential quench protocol and may be linked to the existence of a kink in $g(t)$ at $t = \tau$. We notice similar oscillatory behaviour in the sine and piecewise quenches. In the inset we see the large-$\tau$ behaviour of the stationary magnetisation which is similar to the large-$\tau$ behaviour of the total energy. The deviation from the adiabatic value for quenches with a kink behaves as $|M_s^z - M_a^z| \propto \tau^{-2}$. In contrast, there is a $|M_s^z - M_a^z| \propto \tau^{-4}$ behaviour in quenches without such kinks. The same type of approaches are observed for quenches within the paramagnetic regime. On the other hand, for quenches through the phase transition, no oscillations are observed and the approach to the adiabatic limit is much slower, ie, $|M_s^z - M_a^z| \propto \tau^{-1/2}$ (right panel).

The relaxation to the stationary value is described by the time-dependent part of the magnetisation ($t > \tau$)

$$M_r^z(t) = M^z(t) - M_s^z = -\frac{2}{\pi} \int_{-\pi}^{\pi} dk \, \mathrm{Re} \left[ f(k) e^{2i\omega_k t} \right], \tag{61}$$

where we recall that $\omega_k = \varepsilon_k(g_f)$ defines the single-mode energies after the quench and we

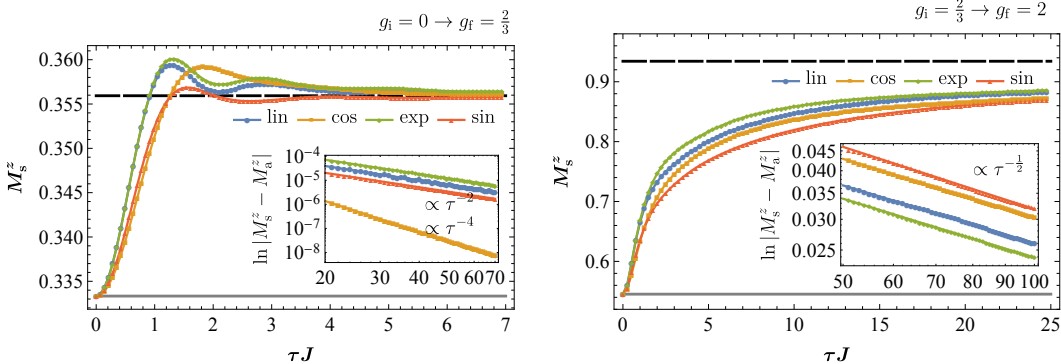

Figure 5: Stationary part of the transverse magnetisation as a function of the quench duration for several quench protocols. The dashed black and full grey lines show the adiabatic and sudden values respectively. The insets show large $\tau$ behaviour, where the adiabatic value is defined by $M_a^z = \lim_{\tau \to \infty} M_s^z$.

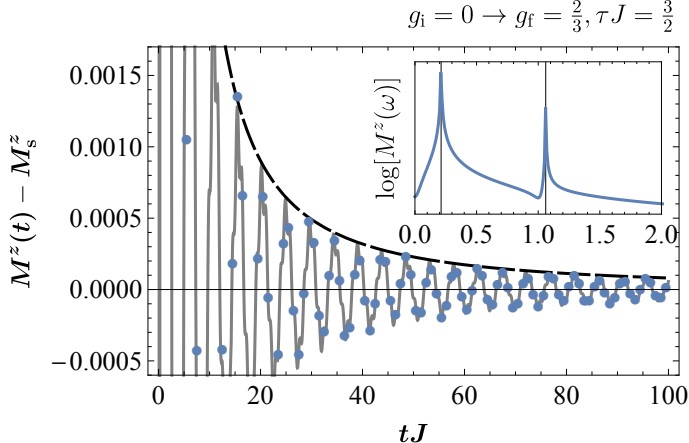

Figure 6: Approach to the stationary value of the transverse magnetisation following a linear quench. The full grey line is the stationary phase approximation result, the dashed line is $t^{-3/2}$ with a constant prefactor, and the blue dots show the numerical evaluation for certain times. The inset shows the spectral analysis of the oscillations demonstrating peaks at frequencies $2\omega_0$ and $2\omega_\pi$ indicated by the vertical lines.

have defined

$$f(k) = \frac{1}{4}e^{-2i\omega_k\tau}\left[\sin^2\theta_k^f\Big(|v_k(\tau)|^2 - |u_k(\tau)|^2\Big) + (\cos\theta_k^f - 1)\sin\theta_k^f\Big(u_k(\tau)v_k(\tau) + u_k^*(\tau)v_k^*(\tau)\Big)\right]. \tag{62}$$

Using a stationary phase approximation we can evaluate the late-time behaviour of this integral to be

$$M_r^z(t) = -\sqrt{\frac{2}{\pi}}\sum_{k_0}|\Phi''(k_0)|^{-3/2}\text{Re}\left[f''(k_0)\exp\left(i\Phi(k_0)t + i\,\text{Sgn}(\Phi''(k_0))\frac{3\pi}{4}\right)\right]t^{-3/2}, \tag{63}$$

where $\Phi(k) = 2\omega_k = 4J\sqrt{1 + g_f^2 - 2g_f\cos k}$ and the stationary points are $k_0 = -\pi, 0, \pi$ respectively. Fig. 6 shows the relaxation of the magnetisation. As in the case of a sudden quench [28, 51] the relaxation follows a $t^{-3/2}$ law. Superimposed on the decay are oscillations with frequencies $2\omega_0$ and $2\omega_\pi$ originating from the stationary points of the phase. The

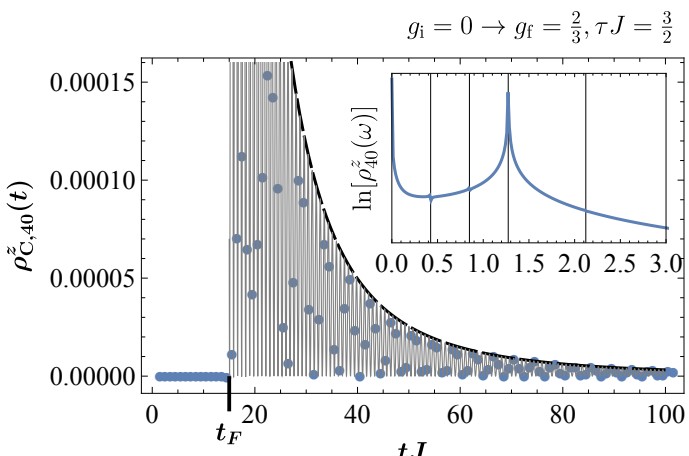

Figure 7: Connected two-point correlation function in the transverse direction for a linear quench. The full line is the stationary phase approximation result, the dashed line is the $t^{-3}$ envelope. The time scale at which correlations set in is indicated by $t_\mathrm{F}$. Inset: Spectral analysis of the oscillations demonstrating a pronounced peak at $2(\omega_0 + \omega_\pi)$ and washed out peaks at the frequencies $2(\omega_0 - \omega_\pi)$, $4\omega_0$, $4\omega_\pi$ respectively.

quench protocol and the duration of the quench implicitly enter the expression of the prefactor of $t^{-3/2}$ via the Bogoliubov coefficients in $f(k)$, while the qualitative behaviour, ie, the power-law decay and oscillations, are unaffected by the details of the protocol.

## 5.3 Transverse two-point function

The two-point function in the transverse direction is given by the Pfaffian (24) which can be evaluated from a $4 \times 4$ matrix. The elements of this matrix are $\alpha_{ij}$ and $\beta_{ij}$, the quadratic correlators introduced in (26) and (27) respectively. At late times we evaluate the behaviour of these correlators using a stationary phase approximation with the result

$$\alpha_{i,i+n}(t) = \alpha_{i,i+n}^{\mathrm{s}} + F_n^1(t)t^{-3/2}, \quad \beta_{i,i+n}(t) = \beta_{i,i+n}^{\mathrm{s}} + F_n^2(t)t^{-3/2}. \tag{64}$$

The stationary parts of the functions are given in (35) and (36), they are found to be negligibly small in comparison to the amplitudes of the time-dependent parts. The prefactors $F_n^1(t)$ and $F_n^2(t)$ are sums of oscillatory terms at $k_0 = -\pi, 0, \pi$, with constant amplitudes and frequencies $2\omega_{k_0}$. Based on this, the connected two-point function in the transverse direction behaves as

$$\rho_{\mathrm{C},n}^z(t) = \langle \sigma_i^z(t)\sigma_{i+n}^z(t)\rangle - \langle \sigma_i^z(t)\rangle^2 \tag{65}$$

$$= 4\left(|\beta_{i,i+n}(t)|^2 - |\alpha_{i,i+n}(t)|^2\right) = \rho_{\mathrm{s},n}^z + G_n^1(t)t^{-3/2} + G_n^2(t)t^{-3}. \tag{66}$$

Since $\alpha_{ij}^{\mathrm{s}}$ and $\beta_{ij}^{\mathrm{s}}$ are negligibly small, the first two terms in (66) are suppressed, and the observed late-time behaviour is a $t^{-3}$ decay. The prefactor $G_n^2(t)$ is a sum of oscillatory terms with constant amplitudes and frequencies $2(\omega_0 + \omega_\pi)$, $2(\omega_0 - \omega_\pi)$, $4\omega_0$ and $4\omega_\pi$. This is shown in Fig. 7. The power-law decay is independent of the quench details, ie, the quench protocol or whether the initial and final values of the quench parameter are in the paramagnetic or the ferromagnetic phase.

The connected two-point function is exponentially small in the spatial separation $n$ up to the Fermi time $t_\mathrm{F}$ when it exhibits the onset of correlations. At later times it shows an algebraic decay $\propto t^{-3}$ with oscillations as shown in Fig. 7. The appearance of the time $t_\mathrm{F}$ corresponds to the physical picture of quasiparticles spreading through the system after

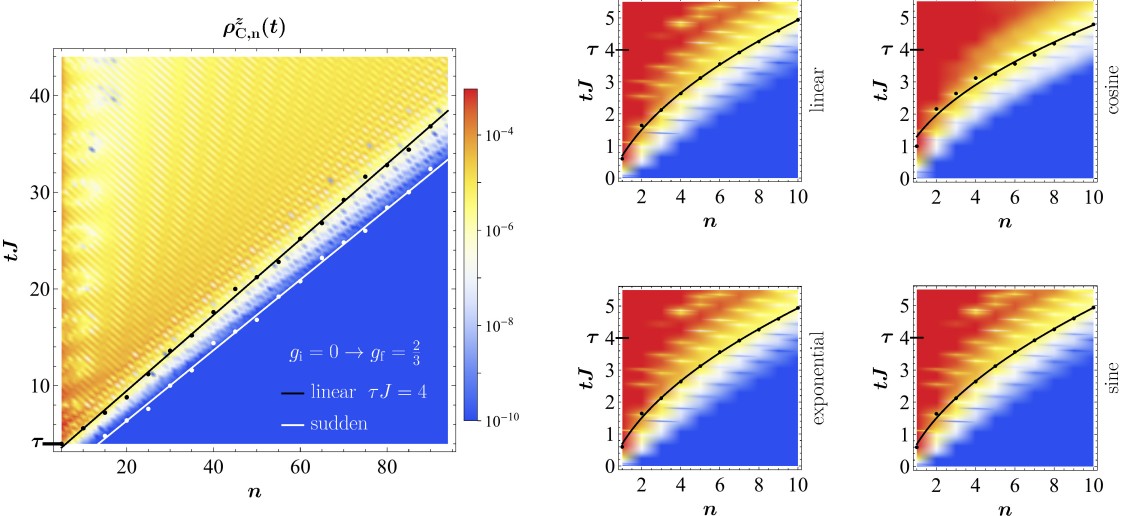

Figure 8: Left: Density plot of the two-point correlation function in the transverse direction following a linear quench. Points of onset are extracted from the first variation with an absolute value not smaller than 1% of the global maximum, lines are linear fits on those points. In white we also indicate the horizon in the sudden-quench case. Right: Density plots of the two-point correlation function in the transverse direction during and after several quench protocols with the same initial and final parameters as on the left. Lines are square root fits on onset points.

sudden quantum quenches as originally put forward by Calabrese and Cardy [10, 11]. The picture adapted to the case of finite-time quenches is as follows [22, 24]: during the quench, $0 \leq t \leq \tau$, pairs of quasiparticles with momenta $-k$ and $k$ are created. The quasiparticles originating from closely separated points are entangled and propagate through the system semi-classically with the instantaneous velocity $v_k(t)$, which is the propagation velocity of the elementary excitations $v_k(t) = d\varepsilon_k[g(t)]/dk$ for a given transverse field $g(t)$. A consequence of this is the light-cone effect—entangled quasiparticles arriving at the same time at points separated by $n$ induce correlations between local observables at these points. This can be seen in Fig. 7, where the connected transverse correlation function does not change significantly until $t_F J \simeq nJ/2v_{max} = 15$. In this rough estimate, we use that the velocity of the fastest mode after the quench is $v_{max}(t) = 2J \min[1, g(t)]$. As stated before, following the onset, the correlations algebraically decay to time-independent values.

The main effects of the finite quench time on the light-cone effect are shown in Fig. 8. Firstly, the quasiparticles are not only created at $t = 0$, but over the entire quench duration $\tau$. Secondly, during the quench, the particles with momentum $k$ propagate with the instantaneous velocity $v_k(t)$, leading to a bending [22, 24] of the light cone for times $t \leq \tau$ clearly visible the plots. Together, these two effects result in a delay of the light cone as compared to the sudden case. A simple estimate for this delay can be obtained by considering the fastest mode created at $t = 0$, which will have propagated at $t = \tau$ to $x_{est} = \int_0^\tau dt\, v_{max}(t)$. On the other hand, in the sudden case the horizon will be at the position $x_{sq} = v_{max}(\tau)\tau$, implying for the delay $\Delta x \approx x_{sq} - x_{est}$, which is consistent with the results shown in Fig. 8.

## 5.4 Longitudinal two-point function

The two-point function in the longitudinal direction can be evaluated from the Pfaffian (25). The corresponding antisymmetric matrix is of dimension $2n \times 2n$, where $n$ is the separation of the spins we are considering. The elements of the matrix are $\alpha_{ij}$ and $\beta_{ij}$ from equations (26)

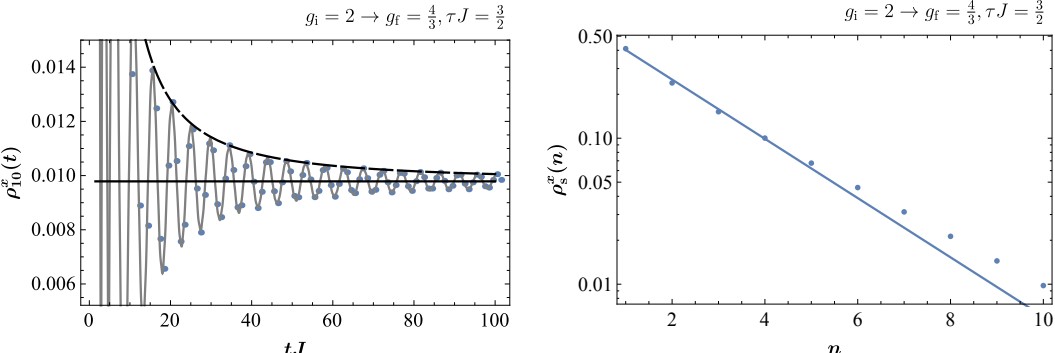

Figure 9: Left: Two-point correlation function in the longitudinal direction following a linear quench. The full line is the stationary phase approximation result, the dashed line its $t^{-3/2}$ envelope. Right: Stationary value of the two-point function for varying spin separations. The full line is an exponential fit to the data.

and (27).

We consider the longitudinal two-point function in the disordered phase only, which equals the connected correlation function because the expectation value of the order parameter vanishes. We analyse its behaviour by using the results of the stationary phase approximation given in (64). Based on this, the connected two-point function in the longitudinal direction and for a quench within the paramagnetic phase behaves as

$$\rho_n^x(t) = \rho_{s,n}^x + F_n(t)t^{-3/2}, \tag{67}$$

in leading order. We note that the power-law decay $\propto t^{-3/2}$ is identical to the sudden-quench case [32]. The prefactor $F_n(t)$ is a sum of oscillatory terms, and $\rho_{s,n}^x$ is exponentially small in the separation $n$, as can be seen in Fig. 9. Similar to the transverse two-point function discussed in the previous section there is a clear light-cone effect with a bending of the horizon during the quench.

Finally we note that the longitudinal two-point function and order parameter have been investigated in the late-time limit after linear ramps within the ferromagnetic phase by Maraga *et al.* [41]. In particular, they showed that the stationary longitudinal two-point function decays exponentially in the separation $n$, ie, $\rho_{s,n}^x \propto e^{-n/\xi}$, with the correlation length $\xi$ being finite even for arbitrarily small quench rates $v_g = (g_f - g_i)/\tau$, implying the absence of order $\lim_{n\to\infty} \rho_{s,n}^x = 0$ after linear ramps. The decay towards this stationary state was not investigated in detail, but in analogy to the sudden-quench case [32] we expect the stationary value to be approached as $\propto t^{-3}$.

## 5.5 Loschmidt echo

It was observed previously [52] that the time evolution of the Loschmidt amplitude after sudden quenches will show non-analytic behaviour provided the quench connected different equilibrium phases. Due to the formal similarity of this behaviour and equilibrium phase transitions this was dubbed dynamical phase transition. Subsequently various aspects of these dynamical phase transitions have been investigated theoretically in various models [53–59], in particular revealing important differences to the usual equilibrium phase transitions [60, 61]. The experimental observation of a dynamical phase transition in the time evolution of a fermionic quantum gas has been recently reported in Ref. [62].

In the present work we investigate the signature of the dynamical phase transition following finite-time quenches in the TFI chain. More precisely, we consider the return amplitude

between the time evolved state $|\Psi(t)\rangle = U(t)|\Psi_0\rangle$ and the initial state $|\Psi_0\rangle = |\Psi(t=0)\rangle$, ie,

$$G(t) = \langle\Psi_0|\Psi(t)\rangle = \langle\Psi_0|U(t)|\Psi_0\rangle. \tag{68}$$

The expectation is that the corresponding rate function $l(t) = -\frac{1}{N}\ln|G(t)|^2$ will show non-analytic behaviour at specific times $t_n^\star$ provided the finite-time quench crossed the quantum phase transition at $g = 1$. We note in passing that the Loschmidt echo after finite-time quenches has been considered previously by Sharma et al. [63]. However, this work considered solely the evolution after the quench, ie, the amplitude $\langle\Psi(\tau)|\Psi(t > \tau)\rangle$, and the finite-time quench appears as a way to prepare the "initial state" $|\Psi(\tau)\rangle$. We stress that, in contrast, we consider the full time evolution both during and after the quench.

To compute the return amplitude (68), we start by noting that the Hamiltonian has the form $H(t) = \sum_{k>0} H_k(t)$ with

$$H_k(t) = A_k(t)\left(c_k^\dagger c_k + c_{-k}^\dagger c_{-k}\right) - iB_k\left(c_{-k}^\dagger c_k^\dagger + c_{-k}c_k\right), \tag{69}$$

ie, the individual Hamiltonians $H_k(t)$ couple only pairs of modes $-k$ and $k$. The time-evolution operator thus also decomposes as $U(t) = \prod_{k>0} U_k(t)$. Next, we revert to the pre-quench operators $\eta_k$ to write the single-mode Hamiltonian (69) in terms of the operators

$$K_k^+ = \eta_k^\dagger \eta_{-k}^\dagger, \quad K_k^- = \eta_k \eta_{-k}, \quad K_k^0 = \frac{1}{2}\left(\eta_k^\dagger \eta_k - \eta_{-k}\eta_{-k}^\dagger\right), \tag{70}$$

which satisfy the SU(1,1) algebra $[K_k^-, K_p^+] = 2\delta_{kp}K_k^0$, $[K_k^0, K_p^\pm] = \pm\delta_{kp}K_k^\pm$. Now we can make the following ansatz for the time-evolution operator [64, 65]

$$U_k(t) = \exp\left[i\varphi_k(t)\right]\exp\left[a_k^+(t)K_k^+\right]\exp\left[a_k^0(t)K_k^0\right]\exp\left[a_k^-(t)K_k^-\right]. \tag{71}$$

From $i\partial_t U_k(t) = H_k(t)U_k(t)$ we then obtain differential equations for the coefficients $\varphi_k(t)$, $a_k^+(t)$, $a_k^0(t)$ and $a_k^-(t)$ which we solve with the initial conditions $\varphi_k(0) = a_k^+(0) = a_k^0(0) = a_k^-(0) = 0$. With this result the return amplitude becomes

$$G(t) = \exp\left[-\frac{iN}{2\pi}\int_0^\pi dk \int_0^t dt' A_k(t')\right]\exp\left[\frac{N}{2\pi}\int_0^\pi dk \ln\left(u_k^i u_k^*(t) + v_k^i v_k(t)\right)\right]. \tag{72}$$

The corresponding rate function is given by

$$l(t) = -\frac{1}{\pi}\int_0^\pi dk \ln\left|u_k^i u_k^*(t) + v_k^i v_k(t)\right|, \tag{73}$$

which we plot in Fig. 10 during and after quenches across the critical point with different quench durations and protocols. The rate function clearly features non-analyticities at times $t_n^\star(\tau)$. We note that the superscript $\star$ denotes the critical times rather than complex conjugation, which we denote by the superscript $*$ throughout the paper. In contrast, we did not observe such non-analyticities for quenches within a phase.

We observe that a short quench reproduces the sudden-quench result [52, 66], whereas for longer quenches there is an offset in the characteristic times. This can be further investigated by considering the return amplitude (72) after the quench $t > \tau$ using the post-quench solutions from (15) for $u_k(t)$ and $v_k(t)$. In this case the corresponding rate function becomes

$$l(t) = -\frac{1}{\pi}\int_0^\pi dk \ln\left|u_k^i c_4^{u*} - v_k^i c_4^{v*} + (u_k^i c_3^{u*} - v_k^i c_3^{v*})e^{-2i\omega_k t}\right|, \tag{74}$$

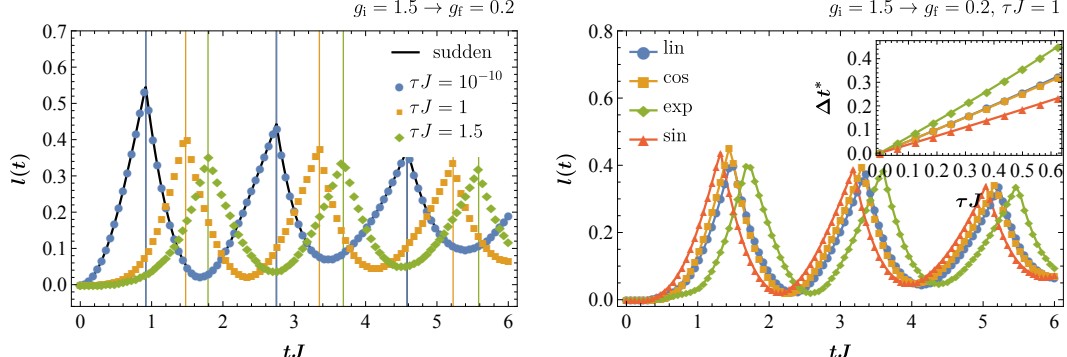

Figure 10: Left: Rate function of the return probability following linear quenches of different durations. Vertical lines are the times $t_n^\star(\tau)$. The full line is the rate function for the sudden quench [52, 66]. Right: Rate function following quenches of various protocols. Inset: Dependence of the offset $\Delta t^\star(\tau)$ on the quench duration and protocol. Full lines are guides to the eye.

where $u_k^i$ and $v_k^i$ are the Bogoliubov coefficients of the initial Hamiltonian and $c_{3/4}^{u/v}$ are the momentum- and quench duration-dependent functions given in equations (18)–(17). When considering the analytic continuation $t \to -iz$ of (74), the argument of the logarithm will vanish at lines in the complex plane parametrised by the momentum $k$ and explicitly located at

$$z_m(k) = \frac{1}{2\omega_k}\left(\ln\frac{u_k^i c_3^{u*} - v_k^i c_3^{v*}}{u_k^i c_4^{u*} - v_k^i c_4^{v*}} + i\pi(2m+1)\right), \tag{75}$$

with $m$ being an integer. The lines (75) will cut the real time axis provided there exists a momentum $k^\star$ with $\mathrm{Re}\, z_m(k^\star) = 0$. The corresponding critical times at which the rate function $l(t)$ will show non-analytic behaviour are given by $t_m^\star = -i\,\mathrm{Im}\, z_m(k^\star)$ with the explicit result

$$t_m^\star(\tau) = \Delta t^\star(\tau) + t^\star\left(m + \frac{1}{2}\right), \quad m = 0, 1, 2, \ldots, \tag{76}$$

where the periodicity is given by $t^\star = \pi/\omega_{k^\star}$ and the offset reads

$$\Delta t^\star(\tau) = \frac{1}{2\omega_k}\arg\frac{u_k^i c_3^{u*} - v_k^i c_3^{v*}}{u_k^i c_4^{u*} - v_k^i c_4^{v*}}\bigg|_{k=k^\star}. \tag{77}$$

We note that $\Delta t^\star(\tau)$ depends on $\tau$ via the coefficients $c_{3/4}^{u/v}$. We also stress that the result (76) is only valid for critical times after the quench $t_m^\star > \tau$. A comparison with the explicit numerical evaluation of the rate function defined via (72) is plotted in Fig. 10; it shows excellent agreement. In particular, the non-analyticities occur periodically and are shifted relative to each other. The latter finding originates from the the fact that the critical mode $k^\star$, obtained from $\mathrm{Re}\, z_m(k^\star) = 0$, depends implicitly on the quench protocol. We also note that the condition $\mathrm{Re}\, z_m(k^\star) = 0$ cannot be satisfied for quenches within the same phase, while for quenches across the critical point such a mode exists.

The analysis above is restricted to $t > \tau$, but for relatively short quenches it captures all critical times $t_m^\star$, since they all occur after the quench. However, for slower quenches kinks in the rate function occur during the quench. We plot several such situations in which $t_1^\star < \tau$ in Fig. 11. In this case, the analysis of the rate function in (73) would require the use of solutions of the differential equations with time-dependent coefficients in (13), which are not always explicitly known.

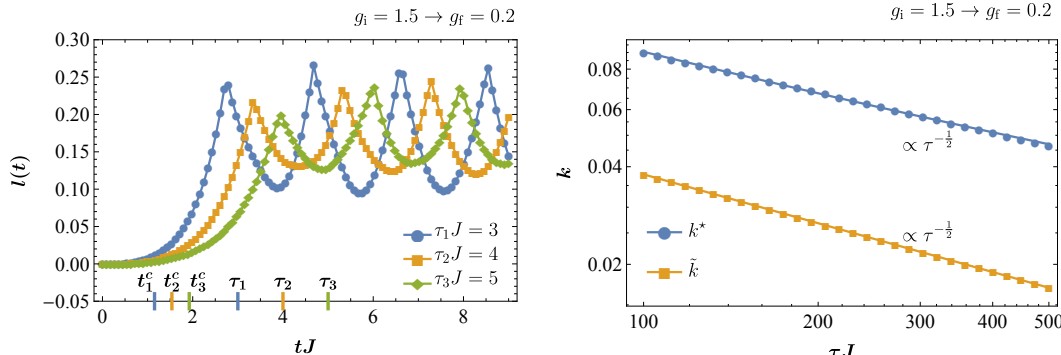

Figure 11: Left: Rate function of the return probability following linear quenches of different durations. We stress that the first critical time $t_1^\star$ occurs during quench, ie, $t_1^\star < \tau$. On the time axis we indicate the quench durations $\tau_i$ as well as the times $t_i^c$ at which the critical point $g_c = 1$ is crossed. Right: Scaling of the critical momenta defined via $\mathrm{Re}\, z_n(k^\star) = 0$ and $n_{\tilde{k}} = 1/2$ for a linear quench. The behaviour is consistent with $k^\star, \tilde{k} \propto \tau^{-1/2}$.

Finally we compare the critical mode $k^\star$ obtained from $\mathrm{Re}\, z_m(k^\star) = 0$ with the mode $\tilde{k}$ defined by $n_{\tilde{k}} = 1/2$, ie, corresponding to infinite temperatures. For dynamical phase transitions after sudden quenches it was found that these two modes are identical [52]. In contrast, for the finite-time quenches we considered here this is not the case, ie, in general we find $k^\star \neq \tilde{k}$. Nevertheless, as shown in Fig. 11 the scaling behaviour of these two critical momenta is consistent with $k^\star, \tilde{k} \propto \tau^{-1/2}$ as expected in the Kibble–Zurek scaling limit [40,67].

## 6 Scaling limit

As is well known, the vicinity of the quantum phase transition at $g = 1$ can be described by the scaling limit [68,69]

$$ J \to \infty, \quad g \to 1, \quad a_0 \to 0, \tag{78} $$

where $a_0$ denotes the lattice spacing, while keeping fixed both the gap $\Delta$ and the velocity $v$ defined by

$$ 2J|1 - g| = \Delta, \quad 2Ja_0 = v. \tag{79} $$

The Hamiltonian in the scaling limit reads

$$ H = \int_{-\infty}^{\infty} \frac{\mathrm{d}x}{2\pi} \left[ \frac{iv}{2}(\psi \partial_x \psi - \bar{\psi} \partial_x \bar{\psi}) - i\Delta \bar{\psi}\psi \right], \tag{80} $$

where $\psi$ and $\bar{\psi}$ are right- and left-moving components of a Majorana fermion possessing the relativistic dispersion relation $\varepsilon(k) = \sqrt{\Delta^2 + (vk)^2}$. Thus we see that the finite-time quenches considered in this article will lead to a time-dependent fermion mass [70] $\Delta(t) = 2J|1 - g(t)|$. We expect our results to directly carry over to the field theory (80), eg, the post-quench relaxation of the transverse magnetisation should follow $M_r^z(t) \propto t^{-3/2}$ as is observed after sudden quenches [51,71].

## 7 Quantum field on curved spacetime

Recently Neuenhahn and Marquardt [42] put forward the idea of using one-dimensional bosonic condensates with time-dependent Hamiltonians in order to simulate a 1+1-dimensional ex-

panding universe. In the following we argue that a similar construction can be performed for the Ising field theory (80). We start from the corresponding action in Minkowski space

$$S_{\text{IFT}} = \frac{1}{2} \int dt\, dx \left[ i v\, \bar{\Psi}\gamma^\mu \partial_\mu \Psi + i\Delta(t)\bar{\Psi}\gamma_3 \Psi \right], \tag{81}$$

where we introduced the two-spinor $\Psi$ and two-dimensional gamma matrices in the Weyl representation via

$$\Psi = \begin{pmatrix} \psi \\ \bar{\psi} \end{pmatrix},\ \bar{\Psi} = \Psi^\dagger \gamma^0 = (\bar{\psi}, \psi),\ \gamma^0 = \sigma^x,\ \gamma^1 = i\sigma^y,\ \gamma_3 = \sigma^z, \tag{82}$$

and set $\partial_0 = \partial_t$ and $\partial_1 = \partial_x$.

On the other hand, the action of a Dirac field with mass $m$ in curved spacetime in 1+1-dimensions is given by [72,73]

$$S_g = \frac{1}{2} \int d^2x\, \sqrt{-g} \left[ i v\, \bar{\Psi}\gamma^a e_a^\mu \nabla_\mu \Psi + i m\bar{\Psi}\gamma_3 \Psi \right], \tag{83}$$

where $g$ is the determinant of the metric tensor, $e_a^\mu$ is the corresponding zweibein and $\nabla_\mu$ denotes the covariant derivative. Specifically we consider the spatially flat Friedmann–Robertson–Walker metric

$$ds^2 = dt^2 - R^2(t)dx^2, \tag{84}$$

which describes a homogeneous, spatially expanding spacetime. With conformal time $d\eta = dt/R(t)$ the metric becomes

$$ds^2 = R^2(\eta)\left( d\eta^2 - dx^2 \right) = R^2(\eta)\eta_{\mu\nu}dx^\mu dx^\nu = g_{\mu\nu}dx^\mu dx^\nu, \tag{85}$$

where $\eta_{\mu\nu} = \text{diag}(1,-1)$ is the Minkowski metric. Using this, the zweibein defined via $\eta_{ab} = e_a^\mu e_b^\nu g_{\mu\nu}$ is found to be $e_a^\mu = R^{-1}\delta_a^\mu$ with the inverse $e_\mu^a = R\delta_\mu^a$. The covariant derivative is given by

$$\nabla_\mu = \partial_\mu + \frac{1}{8}\omega_\mu^{ab}[\gamma_a, \gamma_b] = \partial_\mu - \frac{\partial_\eta R}{2R}\delta_\mu^1 \gamma_3, \tag{86}$$

where we have evaluated the spin connection $\omega_\mu^{ab} = \eta^{bc}\omega_\mu{}^a{}_c$ defined using the Christoffel symbols $\Gamma^\lambda_{\mu\nu}$ as $\omega_\mu{}^a{}_b = -e_b^\nu(\partial_\mu e_\nu^a - \Gamma^\lambda_{\mu\nu}e_\lambda^a)$. Thus with $-g = R^4$ we find

$$S_g = \frac{1}{2} \int d\eta\, dx\, R \left[ i v\, \bar{\Psi}\gamma^\mu \partial_\mu \Psi + i\bar{\Psi}\left( mR\gamma_3 + \frac{v\partial_\eta R}{2R}\gamma^0 \right)\Psi \right]. \tag{87}$$

Finally, rescaling the fields according to $\chi = \sqrt{R}\Psi$, we obtain

$$S_g = \frac{1}{2} \int d\eta\, dx \left[ i v\, \bar{\chi}\gamma^\mu \partial_\mu \chi + i m R(\eta)\bar{\chi}\gamma_3 \chi \right], \tag{88}$$

thus establishing the relation $\Delta(t) = mR(\eta(t))$ between the time-dependent gap and the scaling factor in the Friedmann–Robertson–Walker metric. Hence we conclude that the spreading of correlations during the finite-time quench can be interpreted as propagation of particles in an expanding space time. This result is very similar to the relation obtained in the bosonic case [42].

# 8    Conclusion

In conclusion, we have investigated finite-time quantum quenches in the transverse-field Ising chain, ie, continuous changes in the transverse field over a finite time $\tau$. We discussed the general treatment of such time-dependent quenches in the TFI model and applied this framework to several quench protocols. The precise forms of these protocols were chosen to cover different features like kinks in the time dependence. Specifically we derived exact expressions for the time evolution of the system in the case of a linear and an exponential protocol, and for several others we obtained numerical solutions. Furthermore, we constructed the GGE for the post-quench dynamics using the mode occupations of the eigenmodes of the final Hamiltonian.

Using these results, we analysed the behaviour of several observables during and after the quench. Namely, we investigated the behaviour of the total energy, transverse magnetisation, transverse and longitudinal spin correlation functions and the Loschmidt echo. We confirmed that the stationary values to which the observables relax correspond to the GGE expectation values, as was of course expected. The approaches to the stationary values are oscillatory power laws, details of which can be extracted from a stationary-phase approximation. Furthermore, we checked that the stationary values reproduce the corresponding results for sudden quenches in the short-$\tau$ limit as well as the adiabatic expectation values in the long-$\tau$ limit. As a function of the quench time $\tau$ the approach to the adiabatic values was shown to follow different power laws, depending on whether the quench is within a phase, or if it is done across the critical point.

In the time evolution of the two-point functions we observed the light-cone effect known from sudden quenches. In comparison to the sudden case, however, there is an offset in the horizon after the quench as well as a non-linear regime during the quench. These effects can be ascribed to the production of quasiparticles during the quench as well as to the fact that their instantaneous velocities depend on the quench protocol.

Furthermore, we investigated the behaviour of Loschmidt echo and found signatures of dynamical phase transitions when quenching across the critical point, as was observed previously in sudden quenches. We analysed the rate function of the return amplitude and observed smooth behaviour when quenching within a phase, and periodic non-analyticities when quenching across the critical point. The latter are delayed as compared to the sudden-quench case. We found exact analytical expressions for the post-quench times at which these non-analyticities occur, characterising their periodicity and the delay. In addition, we showed numerically that the non-analyticities can occur during the quench as well, provided the quench duration is sufficiently long.

Finally, we looked into the scaling limit of the theory in which the transverse-field quench corresponds to a quench in the mass of the Majorana fermions. We showed that, alternatively, we can describe the quenching procedure by a field theory with constant parameters put on a curved expanding spacetime, as was proposed previously for a bosonic field theory.

In the future it would be interesting to study the behaviour of other models during and after finite-time quenches. As such investigations presumably have to be based on numerical simulations, the results presented here may serve as an ideal starting point. From our perspective, a natural model to begin with would be the axial next-nearest-neighbour Ising chain, which, in the language of Jordan–Wigner fermions, would correspond to an interacting, non-quadratic theory. While universal results like the scaling behaviour close to the phase transition are expected to be identical to the TFI chain, the non-universal details of the time evolution may reveal interesting interaction effects and their interplay with the energy scale set by the finite quench time.

## Acknowledgements

We would like to thank Piotr Chudzinski, Maurizio Fagotti, Nava Gaddam, Markus Heyl, Dante Kennes and especially Michael Kolodrubetz for useful comments and discussions. This work is part of the D-ITP consortium, a program of the Netherlands Organisation for Scientific Research (NWO) that is funded by the Dutch Ministry of Education, Culture and Science (OCW). This work was supported by the Foundation for Fundamental Research on Matter (FOM), which is part of the Netherlands Organisation for Scientific Research (NWO), under 14PR3168.

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
