# Peer review of "Time evolution during and after finite-time quantum quenches in the transverse-field Ising chain"

_SciPost Physics, doi:SciPost Phys. 1, 003 (2016)_

## Round 1 · Referee Report · Anonymous (Referee 1) · 2016-9-17

Strengths

1- well-written and organised, clear presentation 2- extensive analysis of the subject (more than one examples)

Weaknesses

1- motivation not clearly highlighted in the introduction

Report

This paper analyses the effects of a finite quench duration on non-equilibrium dynamics in the transverse-field Ising chain. It presents exact results for the evolution of physical observables using analytical or numerical solution of the equations of motion for several types of protocols and discusses in detail the interplay with various aspects of non-equilibrium physics: the long time steady state and GGE, the crossover from instantaneous to adiabatic limit, relation to Kibble-Zurek scaling laws, horizon effect and the quasi-particle picture, dynamical phase transitions and a connection with gravitation physics in the scaling limit.

Overall the paper is well written and contains an interesting collection of results together with a discussion of their physical significance.

Requested changes

I would propose the following minor optional changes that could further improve the presentation:

1- What is missing from the introduction is a discussion of the motivation for the study of finite-time quenches: what are the questions to be addressed (e.g. effect of quench duration on horizon effect, on critical times of dynamical phase transitions etc.).

2- In the discussion of different scaling behaviour of total energy with the quench duration tau for different protocol types: is there any intuitive explanation of how the presence of kinks in the time protocol affects the scaling?

3- In the discussion of the quasiparticle picture and horizon: quasiparticles with different group velocities are of course created at different times during the quench, not all at t=0, therefore it could be possible that the fastest quasiparticles created during the quench are not the first to arrive at a given point. Does this matter in estimating the horizon time t_F? Or is it not important or unclear to see in the numerics?

4- In the discussion of dynamical phase transitions and critical times before the end of the quench, it would be interesting to: i) indicate in Fig.11 not only the end-time \tau but also the precise time at which the critical value g=1 is crossed, for comparison purposes. ii) comment on any dependence on the type of quench protocol: do such critical values during the quench occur for all four types considered?

---

## Round 1 · Referee Report · Anonymous (Referee 2) · 2016-9-19

Strengths

  1. Scientifically robust results

  2. Carefully prepared and written

  3. Comprehesive analysis of the problem

Weaknesses

1.Results complement the existing literature on the subject.

  1. Different protocols considered do not really lead to different qualitative features

Report

The authors study a time-dependent quench protocol in the transverse Ising chain. In particular, the coupling g(t) to the transverse field is allowed to vary in time in a certain window. Making a self-consistent ansatz for the Jordan Wigner fermions, they solve mostly numerically the Heisenberg equations of motion and compute the one-point function of the transverse magnetization and the longitudinal and transverse two-point functions for several functions g(t). They also discuss the non-analytic features of the returning probability. In my opinion, the paper is carefully written and the conclusions scientifically robust although perhaps not really surprising, in the light of the exsisting literature on the subject. I do recommend the paper and suggest to the authors to consider the following additional remarks.

Requested changes

1)At pag. 12, discussing the asymptotic behaviour of the energy in the system E(tau) for quenches that do not cross the critical point. What is meant with the statement "stationary phase methdos cannot be applied"? I would think that even if n(k=0)=0, stationary phase could be applied and will give rise to a faster decay to E_{gs}.

2)By simple stationary phase, the t^{-3/2} long time behavior of the transverse magnetization is expected in general for any quadratic operator in a quench when the relation between the pre-quench modes and post-quench modes is linear. So eq. (61) and Fig. 6 are actually expected a priori. This in my opinion should be emphasize more clearly.

3)Typo: n->m in eq. 73, see text below.

4)Can the authors briefly remind the physical meaning of the time t*, where the returning amplitude is non-analytic. Is there a way to understand the difference between the different values of t* in Fig. 10 left?

5)The authors may want to consider that derivation of (86) can be slightly simplified observing that in two dimensions the spin-connection drops if we write the Dirac lagrangian in an explicit hermitian form, see for example Nakahara, Geometry, Topology and Physics, IOP 2003 formula (7.229a').

6)Can the authors in their conclusions stress again the main motivation of considering these five quench protocols? Namely, when they show qualitatively different features, or on the other hand observables that are protocol independent (excluding the t^{-3/2} decay of the transvese magnetization)?

---

## Round 2 · Author Response

Dear Editors,

we thank you for forwarding us the referee reports. We are very happy about the positive reports and thank the referees for their constructive suggestions. We have addressed all suggestions and revised the manuscript accordingly.

Please do not hesitate to contact us if you have further suggestions or questions.

Sincerely yours Tatjana Puskarov Dirk Schuricht —————————————— Reply to referee 1

We thank the referee for his/her careful reading and thoughtful report. We have made some changes according to the suggestions. Below we reply to them in detail.

1) We agree with the impression of the referee. We have revised the 1st paragraph of the introduction to improve the motivation. 2) Unfortunately we do not have a good understanding of the differences between protocols with and without kinks. We have revised the text in Sec. 5.1 to clarify our understanding. 3) It should be noted that the quasiparticles will, at least in the quasi-classical picture underlying the discussion, propagate with the instantaneous velocity v(k,t). Thus particles with the same momentum k will possess identical velocities and hence cannot overtake each other. Of course, it may be possible to devise a protocol to create quasiparticles at a specific momentum and time such that these quasiparticles may overtake other quasiparticles created earlier. However, this is not expected for the generic protocols not considered in our manuscript. In any case, our estimate for t_F should only be understood as a rough one. 4.i) We have indicated the times when the QCP is crossed in the revised figure.
4.ii) In Fig. 10(b) we show the Loschmidt echo for different quench protocols while after Eq. (71) we discuss the findings in the text.

—————————————— Reply to referee 2

We thank the referee for his/her careful reading and thoughtful report. We have made some changes according to the suggestions. Below we reply to them in detail.

Concerning the 2. weakness we would like to point out that the different protocols lead to different behaviour in some quantities like the dependence of the total energy on the quench duration, but that one reason to consider the different protocols is also to show that certain features are independent of the protocol and thus show a certain degree of universality. We have revised the 3rd paragraph of the introduction to stress this aspect.

1) Unfortunately we do not have a good understanding of the behaviour of the total energy as function of the quench duration. In fact, our text in Sec. 5.1 is misleading insofar that also for quenches across the QCP a simple stationary phase approximation cannot be applied, since the dependence on the quench time is rather implicit. We have revised the text in Sec. 5.1 to clarify our understanding. 2) We thank the referee for his/her remark. However, we do not fully understand the argument, since also the k-dependence of the coefficients in the transformation between the post- and pre-quench modes will enter the late-time behaviour. 3) We thank the referee for spotting these typos. We have corrected them. 4) Unfortunately we do not have a good physical picture for the meaning of the critical time t^*. 5) We thank the referee for pointing this out. In order to avoid the introduction of the symmetrised derivative we decided to keep the presentation in its current form. 6) We have revised the conclusion to stress again the motivation for considering different quench protocols.

---

## Round 2 · List of Changes

Summary of changes

-Revised 1st and 3rd paragraph of the introduction.
-Added Ref. 49.
-We have revised the text in Sec. 5.1 to clarify our understanding of the behaviour of the total energy as a function of the quench time.
-Clarified at the end of Sec. 5.2 that all qualitative features of the magnetisation are independent of the finite quench duration.
-Clarified that our estimate for the time t_F is only a rough one.
-Added Ref. 58.
-Added a remark on the experimental observation of dynamical phase transitions as well as Ref. 62.
-We corrected the subindices n->m on page 20.
-Replaced Fig. 11(a) to include the times when the QCP is crossed. Revised the caption accordingly.
-Slightly revised the conclusion as suggested by referee 2.

You are currently on this page

Resubmission 1608.05584v2 on 28 September 2016

---

## Editorial Decision

published